# Modeling the three-dimensional connectivity of *in vitro* cortical ensembles coupled to Micro-Electrode Arrays

**Francesca Callegari**[1], **Martina Brofiga**[1,2], **Paolo Massobrio**[1,3]*

**1** Department of Informatics, Bioengineering, Robotics and Systems Engineering (DIBRIS), University of Genova, Genova, Italy, **2** ScreenNeuroPharm, Sanremo, Italy, **3** National Institute for Nuclear Physics (INFN), Genova, Italy

* paolo.massobrio@unige.it

## Abstract

Nowadays, *in vitro* three-dimensional (3D) neuronal networks are becoming a consolidated experimental model to overcome most of the intrinsic limitations of bi-dimensional (2D) assemblies. In the 3D environment, experimental evidence revealed a wider repertoire of activity patterns, characterized by a modulation of the bursting features, than the one observed in 2D cultures. However, it is not totally clear and understood what pushes the neuronal networks towards different dynamical regimes. One possible explanation could be the underlying connectivity, which could involve a larger number of neurons in a 3D rather than a 2D space and could organize following well-defined topological schemes. Driven by experimental findings, achieved by recording 3D cortical networks organized in multi-layered structures coupled to Micro-Electrode Arrays (MEAs), in the present work we developed a large-scale computational network model made up of leaky integrate-and-fire (LIF) neurons to investigate possible structural configurations able to sustain the emerging patterns of electrophysiological activity. In particular, we investigated the role of the number of layers defining a 3D assembly and the spatial distribution of the connections within and among the layers. These configurations give rise to different patterns of activity that could be compared to the ones emerging from real *in vitro* 3D neuronal populations. Our results suggest that the introduction of three-dimensionality induced a global reduction in both firing and bursting rates with respect to 2D models. In addition, we found that there is a minimum number of layers necessary to obtain a change in the dynamics of the network. However, the effects produced by a 3D organization of the cells is somewhat mitigated if a scale-free connectivity is implemented in either one or all the layers of the network. Finally, the best matching of the experimental data is achieved supposing a 3D connectivity organized in structured bundles of links located in different areas of the 2D network.

**Data Availability Statement:** The network model files (Python) and the customized functions (Matlab) used to analyze the data have been deposited in Zenodo. The DOI of the code reported

in this article is https://doi.org/10.5281/zenodo.7331295.

**Funding:** The author(s) received no specific funding for this work.

**Competing interests:** The authors have declared that no competing interests exist.

## Author summary

The brain is the most complex organ of the human body, of which we still have limited knowledge despite the number of extensive studies on this topic. That is because the brain is unique in its structural and functional organization: neurons from different brain areas arranged in a well-defined 3D topology exhibit unique behaviors. To untangle this system, scientists have turned to a multitude of *in vitro* models on chip, which differ in the degree of complexity of the network organization. One of the main aspects that sets apart different models is the introduction of three-dimensionality, which affects greatly the activity of the neuronal population. However, little is known about the organization of these 3D *in vitro* networks due to intrinsic experimental limitations. In the present work, we developed a computational model that is able to reproduce the experimental findings achieved with an *in vitro* model. By matching the activity of our simulated network, whose parameters can be finely controlled, to the activity of *in vitro* 3D cultures, our goal was to infer which kind of 3D connectivity sustains the emerging patterns of electrophysiological activity, therefore matching the experimental findings to the hard-to-observe topological properties.

## Introduction

Nowadays, the use of dissociated neuronal cultures coupled to Micro-Electrode Arrays (MEAs) is a well-established *in vitro* experimental model to explore basic principles of brain functions [1], to investigate their computational properties [2] and to appreciate their electrophysiological modulation when stimulated by electrical [3] or chemical [4] stimuli. Over the years, several attempts have been made to engineer neuronal networks and make them as similar as possible to the micro-structured circuits that characterize the topological organization of the *in vivo* brain ([5] and references therein).

However, it was only in 2008 that a strong limitation of the *in vitro* experimental models based on dissociated cultures was overcome: the brain is intrinsically three-dimensional (3D), and a reduced bi-dimensional (2D) configuration cannot exhibit most of the morphological and electrophysiological key features of the *in vivo* systems [6]. In 2008, Pautot and co-workers developed a protocol to allow a 3D growth of dissociated hippocampal neurons by exploiting a self-assembled scaffold realized with glass microbeads [7]. Some years later, Frega and colleagues combined this protocol with the use of MEAs: the recorded patterns of electrophysiological activity were deeply different from those observed in 2D cultures in terms of both spiking and bursting features [8, 9]. These works paved the way for the use of 3D cultures to better mimic the properties of the *in vivo* brain. Several works dealt with new approaches to increase the cell density in order to be comparable with that found *in vivo*, as well as to improve the mechanical properties (like porosity and stiffness) of the scaffolds to better mimic the extracellular matrix. [10]. 3D electrospun polymers or hydrogels are possible solutions to reach such a goal. Hydrogels ensure low cytotoxicity and favor exchange of gases and nutrients between cells and the surrounding environment. Collagen is another natural soft material that was used as a scaffold to develop 3D neuronal assemblies. In 2015, a model of 3D brain-like tissue was realized by means of silk-collagen proteins [11], where neurons plated in a donut-shaped porous silk sponge were able to develop robust projections within the central region full of collagen, generating dense 3D neuronal networks. The principal drawback of many hydrogel-based materials is their relatively fast mechanical degradation. To overcome this weakness, in 2015, Bosi and colleagues developed a synthetic biocompatible PDMS scaffold

that can be tuned in its micro and nanomechanical properties [12]. Unfortunately, this type of structure hinders the recording of electrophysiological activity. The use of planar MEAs would be, in fact, useless considering that most of the electrodes would be covered by the PDMS. Unfortunately, this is not the only drawback when 3D neuronal networks are coupled to MEAs. Most of the works suffer the intrinsic technological limits of planar devices: only the electrophysiological activity of a very small subset of neurons (localized in the bottom layer and directly coupled to the active area of the MEA) is recorded [8, 9, 13–15]. Only in the last years, new technological efforts were made to develop 3D devices able to map the electrophysiological activity of 3D networks in the 3D space. In 2020, Soscia and co-workers designed an *in vitro* platform to simultaneously measure the electrophysiological activity of three independent 3D cultures [16]. In 2021, Shin and colleagues developed a device able to record the electrophysiological activity of *in vitro* 3D assemblies up to a height of 300 μm [17] and provided a first set of information about the functional connectivity of 3D neuronal assemblies. The authors inferred the 3D functional connectivity by counting the number of links per each electrode and mapping their spatial organization in terms of length of connections. Despite the power of these technological devices that could potentially overcome many of the experimental limitations, topological properties (like clusterization, degree distribution, emerging of small-world properties and modular communities, presence of highly connected neurons (hubs)) of the 3D networks were still not extracted and therefore effective information about the 3D topology is still missing. However, MEAs are not the only technological devices to record the electrophysiological activity of 3D neuronal networks: optical methods are a valid alternative solution to overcome such limitations: in contrast to MEA, optical-imaging-based techniques using dyes or genetically encoded reporter constructs guarantee a higher spatial resolution and recordings at single cell resolution. Among these, calcium-imaging also allows to simplify the analysis of emerging functional activity, avoiding the use of spike sorting algorithms to discern the activity of dense clusters of 3D neurons recorded by planar or 3D electrodes. In addition, the use of optical instruments provides simultaneously information about the interplay between structural connectivity and the emerging activity. This last feature is of extreme interest when we deal with high density (2D and 3D) neuronal assemblies. In 2017, Marom and colleagues developed an optical-accessible *in vitro* model of 3D networks by injecting a genetically encoded calcium indicator [18] to probe the variations of the emerging electrophysiological activity under their natural development. Many works in the literature dealing with *in vitro* cultures exploit functional connectivity hints to infer structural properties [19]. Indeed, functional connectivity is strongly influenced by how it is measured (correlation- vs information-theory based algorithms for example), and some caveats are required to translate functional features into structural ones. First, functional measures are subject to fluctuations if achieved from short periods of recordings. Theoretical studies demonstrated the requirement of at least 10 minutes of recordings to avoid unstable results [20]. In addition, the achieved functional properties of a network can be considered predictors of the structural ones, only at small-scale, i.e., considering small ensembles of neurons [21]. Finally, it is worth mentioning the state-dependency of the functional connectivity: in other words, the same structural connectivity can originate different functional networks which exhibit different topological features. Thus, taking into consideration the weak points of the approach, the correlation between structural and functional connectivity can be considered a good indication on topological properties. In this work, we exploited such interdependency and the structural connectivity rules implemented in the computational model of the 3D cortical networks were inspired by the functional insights.

To the best of our knowledge, only one computational study relative to 3D cultures was performed in 2015 by Bosi and co-workers. The authors developed a large-scale 3D neuronal

network model made up of adaptive exponential integrate-and-fire neurons to elucidate the hidden properties of a 3D *in vitro* network grown on a synthetic polymer-based scaffold with embedded multi walled carbon nanotubes [12]. Due to the mechanical properties of their scaffold, neurons were randomly and uniformly confined in a virtual box to reflect the same cellular distribution of the experiments and connected by means of a Gaussian radial basis function. The authors postulated and tested different variations of the Gaussian connectivity rules for both 2D and 3D configurations. After matching the inter-burst interval values originated by the simulations of the 2D and 3D networks to the ones derived from calcium imaging techniques, the authors characterized the results of their simulations in terms of the topological features [22] of the network. They found a simultaneous increase of the clustering coefficient and no significant variations in the mean path length of the 3D networks, although the average geometrical length of the links increased [12].

Following a reverse engineering approach, we reproduced the dynamics exhibited by 3D cortical networks coupled to MEAs by means of a computational model made up of synaptically connected leaky integrate-and-fire (LIF) neurons. The model is essentially inspired by the Pautot method, where 3D cultures are created with different layers of glass cell-enriched microbeads. It aims at reproducing *in silico* this technique by stacking several layers of neurons and at investigating different possible connectivity motifs among them. In the Pautot and in following works that exploited the same protocol, there were no quantitative measures about how neurons were synaptically connected, and which topological characteristics emerged, both from a structural and a functional point of view. To design the *in silico* model of 3D networks, in the present work we took inspirations from the 2D topological organization (i.e., the one relative to the layer directed coupled to the active area of the MEA), inferred with functional connectivity investigations in dissociated 2D networks [19, 20, 23]. The results achieved with the simulations of the developed network model demonstrated that a 3D environment influences the electrophysiological activity of the network. In particular, we observed that the introduction of a 3D environment (made up of at least four layers, about 200 μm) induced a reduction in both firing and bursting rates.

Considering the experimental evidence of a scale-free functional connectivity observed in *in vitro* cultures, we explored the possibility of a structural implementation of this feature. We investigated the effect brought by the modification of the topological organization with a scale-free structural connectivity, finding that such a topology limits the overall impact of the third dimension on the network. Eventually, the results were compared with those obtained from 3D in vitro cortical cultures, highlighting the relevance and the power of our *in silico* model. Eventually, the results were compared with those obtained from *in vitro* cortical cultures, highlighting the relevance and the power of our *in silico* model.

## Results

In this section, results from the variation of the different constitutive parameters of the 3D neuronal model are presented and compared in order to unravel the contribution of each in the genesis of the spontaneous electrophysiological activity. Generally, the model consists of a spiking network of Leaky-Integrate and Fire (LIF) excitatory and inhibitory neurons arranged in layers with a density comparable to 2D *in vitro* cultures (cf. Materials and Methods). These layers constitute the core element of the model, and they were piled up to create a 3D structure. To create a functional network, different connectivity patterns were implemented, guided by experimental evidence. The reference connectivity both within and between the layers follows a gaussian probability (Fig 1, $G^{2D}$, $G^{3D}$) over distance with a limitation over the maximum number of established connections (cf. Materials and Methods).

| Description | | # sim |
|---|---|---|
| $G^{2D}$ | Gaussian connectivity rule in a single layer | 20 |
| $G^{3D}$ | Gaussian connectivity rule within every layer of the 3D population. | 25 |
| $G_{rnd}^{3D}$ | Gaussian connectivity rule within every layer of the 3D population. The neurons that project to the other layers (source neurons) are a random subset of the total neurons of each layer (Figure 6F) | 4 |
| $G_C^{3D}$ | Gaussian connectivity rule within every layer of the 3D population. The source neurons are a located in a central cluster of each layer (Figure 6G) | 4 |
| $G_{5C}^{3D}$ | Gaussian connectivity rule within every layer of the 3D population. The source neurons are a located in five clusters (Figure 6H) | 4 |
| $SF^{2D}$ | Scale Free connectivity rule in a single layer | 28 |
| $SF_{all}^{3D}$ | Scale Free connectivity rule within every layer of the 3D population. Each neuron could project afferences to the other layers. | 22 |
| $SF_{L0}^{3D}$ | Scale Free connectivity rule within only the readout (bottom, L0) layer of the 3D population. The connections within the other layers follow the Gaussian connectivity rule. Each neuron could project afferences to the other layers. | 12 |

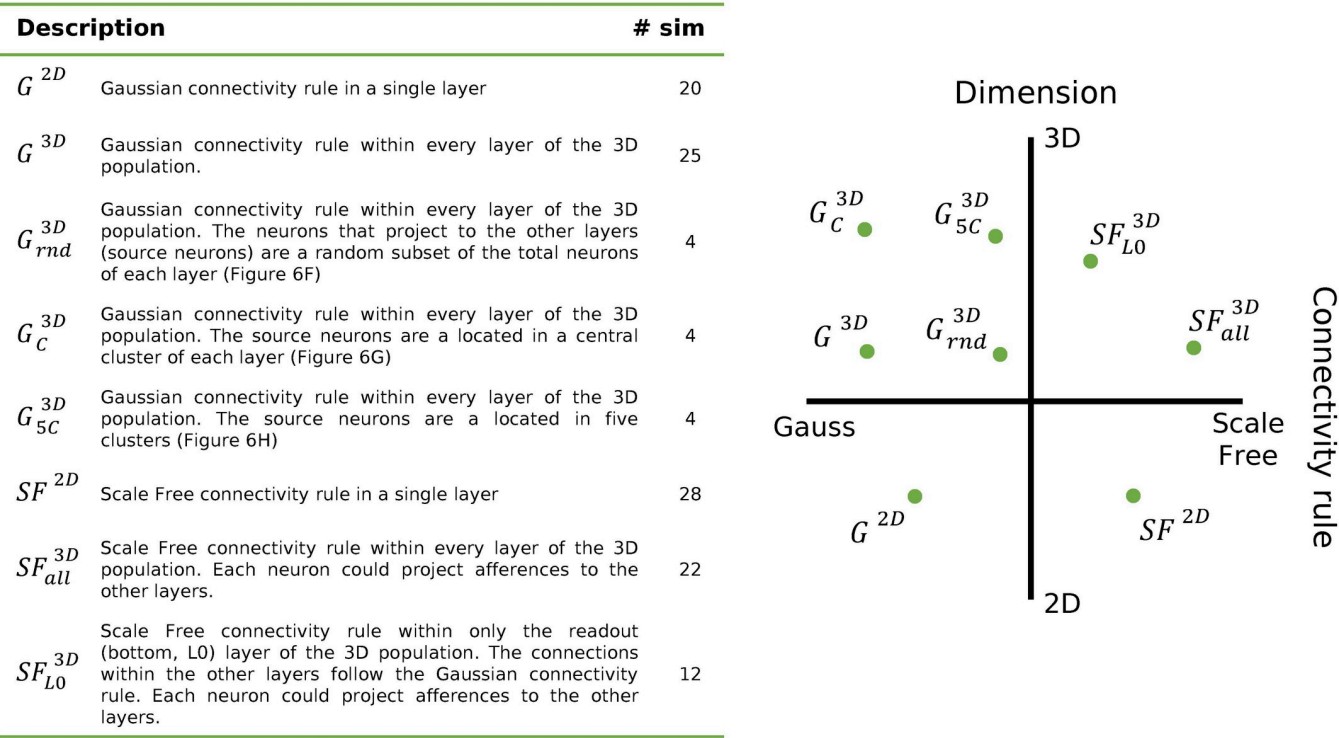

**Fig 1. Representation of the dataset.** All the configurations analyzed in the paper are reported here with their abbreviation and a brief description. For all the 3D, the connectivity among the layers (inter-layer connections) follows a Gaussian distribution scaled over distance as reported in Sect 3D connectivity.

Firstly, we examined the intrinsic dynamics of this reference model and tested the effect of the 3D configuration on the activity of the network by comparing it to 2D structures whose connectivity is ruled by a Gaussian connectivity ($G^{2D}$) as well. Then, we investigated the effect of the number of layers defining the 3D networks to assess which configuration starts producing a significant change in the electrophysiological activity moving from a 2D to a 3D configuration. In this way, we were able to determine the height (i.e., number of layers) at which the model could be functionally considered a 3D structure. Next, the influence of the implemented structural connectivity within each layer was explored, to test whether the functional characteristics observed in MEA recordings underlie a connectivity also in the 3D configuration. In particular, the functional scale-free (SF) connectivity inferred in *in vitro* studies was translated in a structural connectivity fashion either in only the bottom layer (that as of now is the only one that has been observed *in vitro*, Fig 1, $SF_{L0}^{3D}$) or in all the layers of the model (Fig 1, $SF_{all}^{3D}$). A similar investigation was devised for the inter-layer connections with the rationale of inferring the *in vitro* connectivity from the activity patterns recorded only at the readout (bottom) layer. In this case, different positions of the source neurons were conceived, with inter-layer connections generating either in clusters (Fig 1, $G_C^{3D}$ and $G_{5C}^{3D}$) or from random areas (Fig 1, $G_{rnd}^{3D}$). Eventually, the data of the last different configurations were compared to *in vitro* data of 3D cultures recorded by means of MEAs. At the beginning of each section, the conditions of the simulations are summarized for the sake of clarity. Unless otherwise stated, in all the following analyses, the presented data are relative to the activity of the bottom layer (readout layer), to emulate the experimental condition achieved by using planar MEAs.

### Three-dimensionality slows down spiking and bursting activity of cortical networks

Firstly, we tuned the parameters and characterized a 2D network that aimed at mimicking the 2D layout of a 2D *in vitro* cortical culture. This 2D fundamental unit is made up of both excitatory and inhibitory neurons, connected following a Gaussian probability over distance (that is the reference connectivity rule of the present work and was named $G^{2D}$ hereafter) as better described in the Materials and Methods section.

The 3D network of our model was built using the 2D fundamental unit as constitutive element. We tried to mimic the experimental construction of a 3D network with glass microbeads as scaffold. Briefly, as in Pautot's and Tedesco's works [7, 9], the neuronal 3D cultures were created by plating the cells on the glass microbeads, then the cells and beads suspension was transferred to the MEAs, where they self-assemble in stacked layers (details in Sect *In vitro* model). In the computational model, this process was reproduced by piling up different 2D units (details and parameters can be found in the Materials and Methods).

As dataset for the 3D configurations, we considered all the data obtained from simulations where a structural 3D network was created (2 up to 6 layers) with the reference connectivity rules (Fig 1, $G^{3D}$). That means that the connections both within and among the layers are established following a Gaussian probability over distance. The 3D data was compared against 2D networks with the same Gaussian connectivity rule within the layer (Fig 1, $G^{2D}$).

One of the main parameters that influenced the organization of the network activity in terms of random spiking, density of network burst and synchronization of the activity itself was the inhibitory/excitatory ratio. Qualitatively, S1 Fig shows 60 seconds of simulated activity in different 2D neuronal populations that differed only in the percentage of inhibitory cells in the network, starting from 10% up to 50%, built as described in the Materials and Methods section. The inhibitory network by itself was not able to generate a sustained activity, exhibiting only few random spikes, with very low values of Mean Firing Rate (MFR) that, as per our definition, did not quantify as active network. S1A Fig shows an example of a network with 50% of inhibitory neurons, which displayed a very low scattered spiking activity with a MFR value of 0.08 sp/s ($<0.1$ sp/s), and the complete lack of bursts. The activity intensified as the inhibitory/excitatory ratio decreased, reaching a MFR $> 0.1$ sp/s (active node) only when the percentage of inhibitory neurons dropped to 40% S1B Fig). However, the activity started to organize into bursts and network bursts only at 30% inhibitory cells (S1C Fig) and only at 20% (S1E Fig), it showed a MFR comparable to the one found *in vitro* [19, 24], as well as the temporal organization of the network bursts [25]. With lower percentages of inhibitory cells (S1D Fig), the network lost its balance and the firing rate shot at unphysiological values higher than 190 sp/s. Therefore, within the fundamental 2D unit, it was kept the balance of the inhibitory/excitatory neurons at 20%, in line with physiological observations found on dissociated cortical cultures [26].

The result of the chosen combined parameters of these networks (Table 1) gave rise to a sustained spiking and bursting activity also in the 3D configuration, with network bursts that involved almost the entire network, as shown by the 60-s raster plots of a 4-layer network in Fig 2A–2D (and the respective close-ups of a single event in the lower row). Considering the population events, we considered their propagation among the layers. Qualitatively, the IFR traces reported in Fig 2 (lower row) highlight the time stamp of the peak associated with one of the network bursts. From these values, we inferred the delay in the propagation of the event between the layers. In particular, this representative event started from the second layer, propagated to the third and the first ones and only then arrived in the fourth layer, that is the most active. In general, we observed a high variability in the generation and propagation of the network events (as demonstrated by the similarity maps of S2A–S2C Fig, obtained with the

**Table 1. List of the parameters of the 2D model.**

| 2D model parameters | Values |
|---|---|
| Area | 736'164 μm$^2$ |
| Distance between cells | 26 μm |
| Percentage of inhibitory cells | 20% |
| Threshold potential | - 50 mV |
| Reset potential | - 70 mV |
| Refractory period | 5 ms |
| Maximum probability ($p_{MAX}$) | 0.2 |
| Resting potential ($E_L$) | - 70 mV |
| Membrane time constant ($\tau_m$) | 10 ms |
| Leakage conductance ($g_L$) | 10 nS |
| Sub-threshold input ($I_{noise}$) | $g_l \cdot$ {random integer in [0, 5]} $\cdot$ 6.4 mV |

Victor-Purpora statistics [27] described in the S1 File), possibly brought by the intricate connections established in the 3D space. Moreover, we observed that the number of involved neurons during these events significantly decreases if compared to 2D networks (S2D Fig). This fact suggests that the stereotyped activity of the network is on some level severed by the introduction of the third dimension, as a lower percentage of cells is involved in the network event.

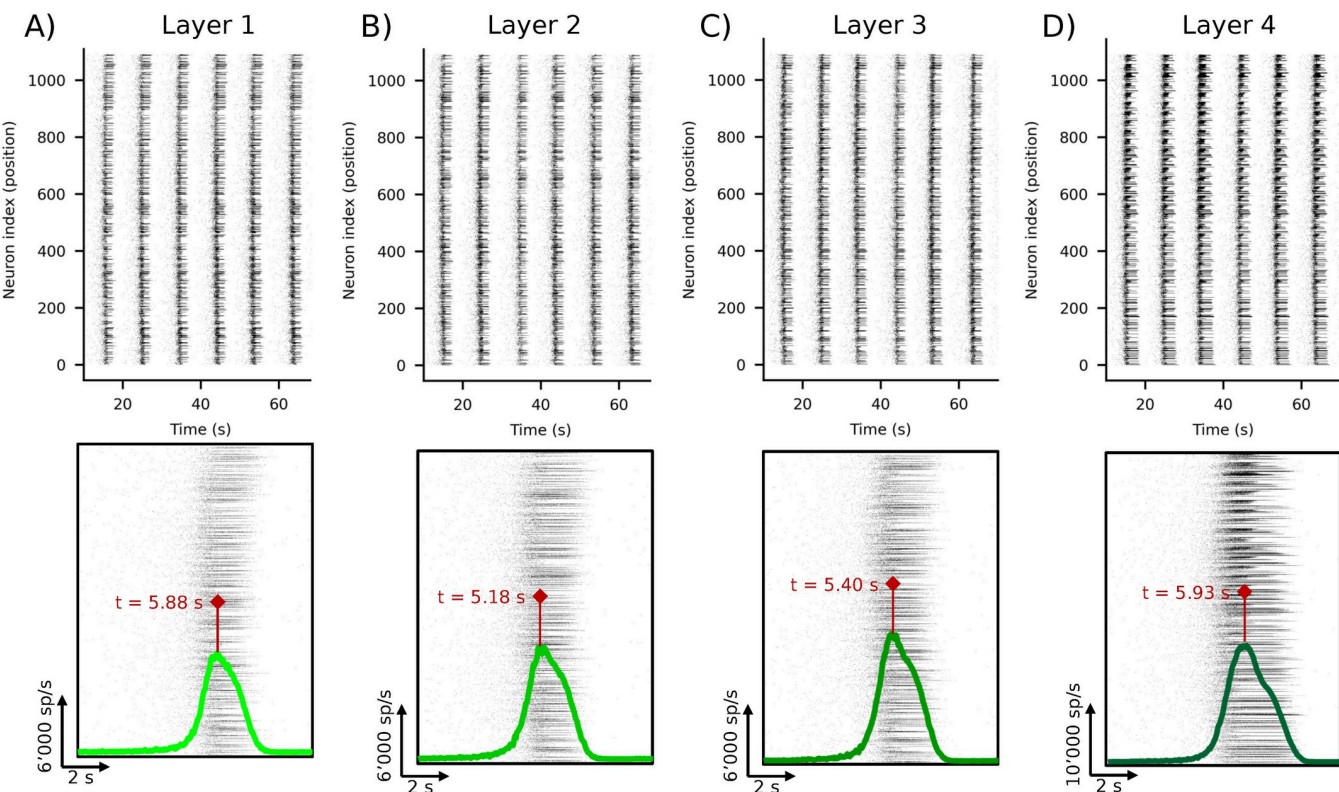

**Fig 2. Raster plots and relative IFR traces of the layers of a single population with Gaussian connectivity ($G^{3D}$) that guarantees in vitro-like patterns of electrophysiological activity where bursting and spiking activity coexist. (First row)** 60-s simulated spontaneous electrophysiological activity of the layers of a representative 4-layer 3D network (each panel **A-D** refers to a single layer of the population, with **A** being the bottom/readout layer). This configuration allows to reproduce values of firing and bursting rates comparable with the experimental recordings of mature cortical cultures as well as the presence of population events (network bursts) involving most of the neurons of the network. **(Second row)** Close-up of a single population event with super-imposed IFR traces (**green**) and temporal stamps of the relative maxima (**red**).

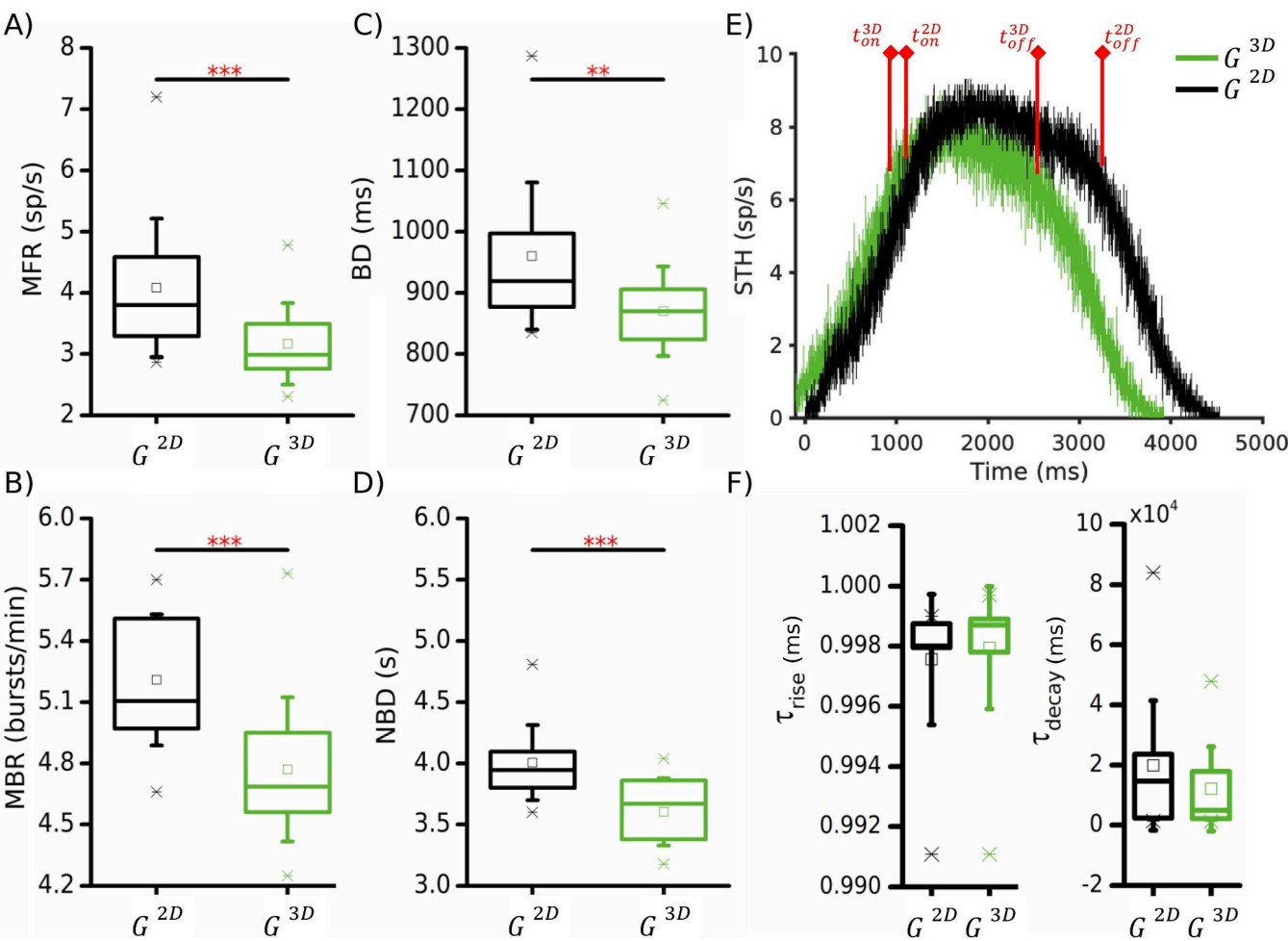

**Fig 3. Electrophysiological features of 2D (black, $G^{2D}$) and readout layer of 3D (green, $G^{3D}$) networks.** (A) Mean firing rate, (B) mean bursting rate, (C) burst duration, (D) network burst duration. (E) Average Spike Times Histogram (STH) of a 2D and a 3D exemplary configuration (bin = 1 ms). $t_{on}$ and $t_{off}$ indicate the temporal instant when the plateau phase of the network burst starts and ends, respectively. (F) Temporal time constants of the (**left**) rise and (**right**) decay phases of the network bursts. In these statistics, we have considered all the simulated structural 3D networks. (* refers to 0.01<p<0.05, ** 0.001<p<0.01, *** p<0.001, Kruskal-Wallis non-parametric test).

However, considering the higher level of complexity in the evaluation of the whole network and the lack of an *in vitro* experimental counterpart (data from planar MEAs), the focus of our analysis was on the examination of the metrics obtained only from the bottom (or readout) layer of 3D networks and from 2D ones to try inferring some differences based solely on this reduced observation point of view.

The fundamental 2D unit (Fig 3, *black*) displayed *in vitro*-like values of firing (4.1 ±1.1 sp/s, Fig 3A), and bursting rate (5.21 ± 0.31 bursts/min, Fig 3B), with bursts lasting 960.0 ±117.4 ms (Fig 3C). At the network level, the 2D simulated networks exhibited on average 6 network bursts per minute, with an average duration of 4.0 ± 0.3 s (Fig 3D, *black line*).

The addition of layers (Fig 3, *green*) brought to a general decrease in the spiking and bursting activity of the network. The MFR and mean bursting rate (MBR) values decreased to 3.3 ± 0.7 sp/s (p = 3.8 · $10^{-4}$, Fig 3A), and to 4.8 ± 0.4 bursts/min (p = 6.7 ·$10^{-5}$, Fig 3B), respectively. A similar trend was found also for the burst duration (BD) dropped to 871.1 ± 71.9 ms when the 3D configuration was considered (p = 0.004, Fig 3C). Generally, the changes of the

average behavior of the single neurons were reflected also in the collective activity of the entire assembly (network bursts, Fig 3E). We observed a decrease in the network burst duration (NBD) to 3.6 ± 0.3 s, (p = 0.003, Fig 3D). No significant variations were observed in the rise (Fig 3F, *left*) and in the decay (Fig 3F, *right*) phases of the network burst, suggesting that the addition of the third dimension itself is not a sufficient condition to modify the recruitment and deactivation time of the collective behavior of the neuronal network, it affects just its duration.

### Influence of the number of layers of the 3D network in spiking and bursting features

From an experimental point of view, a crucial question was to understand how many layers are necessary in order to start producing different patterns of activity with respect to the 2D networks: by means of the model, we evaluated the configuration (in terms of number of stacked layers) that could be functionally and physiologically considered an effective 3D model. Thus, following the experimental procedures related in [8] (3D cultures created with glass microbeads that self-assemble in layers), we incrementally added one layer at a time from 2 up to 6 layers to try to find the threshold for a 3D-like behavior.

Interestingly, the bursting activity was strongly affected by the adding of layers, at both the single neuron and the network level. In fact, both the MBR (4.7 ± 0.3 bursts/min, p = 0.02, Fig 4B) and the NBD (3.6 ± 0.2 s, p = 0.008, Fig 4D) showed a first significant difference with respect to the 2D configuration already when the third layer was added. Instead, a fourth layer

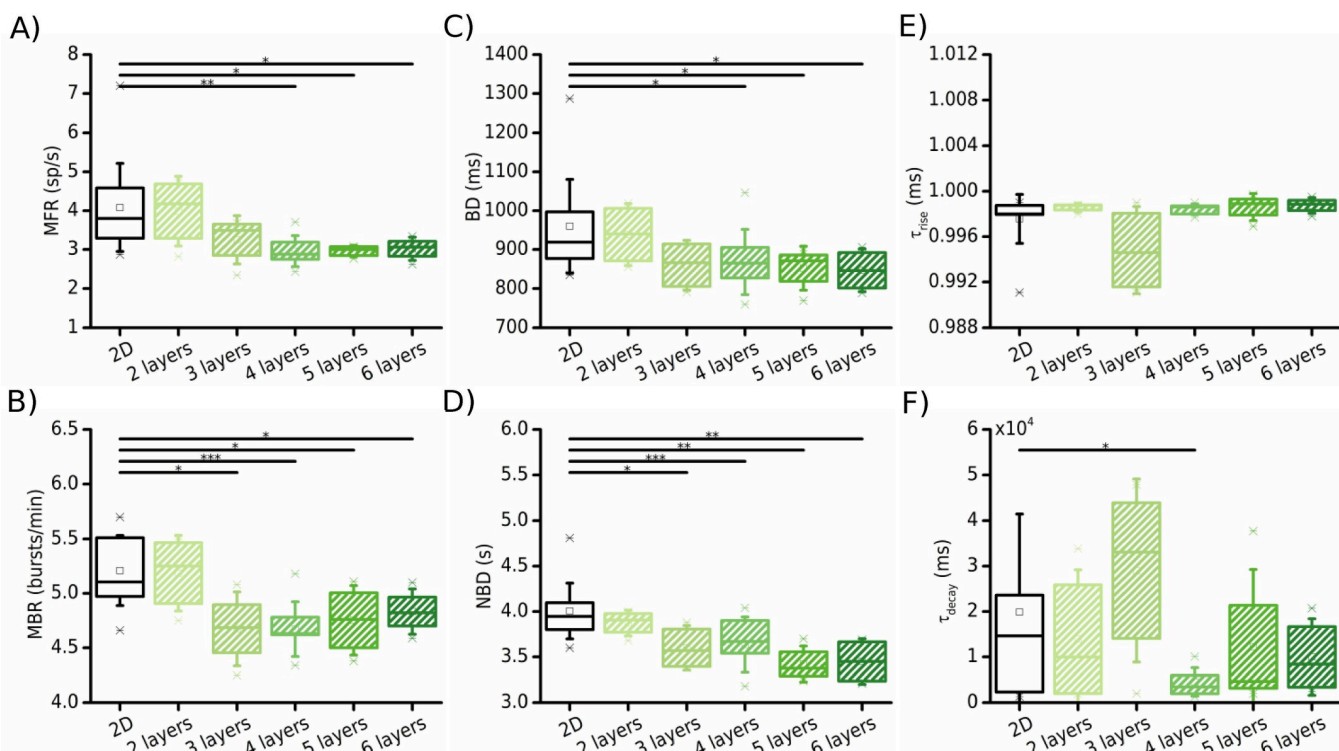

**Fig 4. Effect of the number of layers (swept from 2 to 6) of the 3D model (green boxes) on the electrophysiological activity compared to the 2D networks (black boxes).** (**A**) Mean firing rate, (**B**) mean bursting rate, (**C**) burst duration, (**D**) network burst duration. The temporal time constants of the rise and decay phases of the network bursts are reported in (**E**) and (**F**), respectively. (* refers to $0.01 < p < 0.05$, ** $0.001 < p < 0.01$, *** $p < 0.001$, Kruskal-Wallis non-parametric test).

was necessary to get a significant effect on the MFR (3.09 ± 0.4 sp/s, p = 0.001, Fig 4A) and on the burst duration (868.0 ± 79.2 ms, p = 0.03, Fig 4C). Overall, combining the information coming from the spiking and bursting activity, we observed that 4 layers are necessary to change the dynamical state of the network from the 2D configuration. The addition of further layers (in the model up to 6), did not induce significant differences in the network activity with respect to the first significant variation. As mentioned in the previous section, three-dimensionality itself was not able to produce a change in the rising and decaying phases of the network burst activity. It had a significant role only on the duration of the events themselves, not on the recruitment time (Fig 4E) or in their decay phase (Fig 4F), except in the case with 4 layers, where $\tau_{decay}$ dropped to 4544.3 ± 2948.0 ms (p = 0.02). This observation stood for all the number of layers, which reached a cumulative height of 300 μm along the $z$-axis.

### Effect of the different connectivity rules at the single layer level

From a functional point of view, different types of connectivity features were observed in 2D *in vitro* models [19, 28–30]. Among them, a scale-free (SF) distribution of the functional connectivity was observed both in hippocampal [31] and cortical [20] circuits, indicating the presence of hubs which are suggested to coordinate the activity of the entire circuitry under examination [19, 31]. As a maintenance of the topological features between functional and structural networks can be kept [22], we reproduced the functional features of the SF network in the 3D model (details on the implementation can be found as Rule 2: SF connectivity). In particular, we established a SF connectivity first in each layer of the 3D network (Fig 1, $SF_{all}^{3D}$), then only in the readout or bottom layer (Fig 1, $SF_{L0}^{3D}$). In this case, the connectivity within the other layers was maintained as in the previous case (i.e., the reference connectivity rule where the connections were established following a Gaussian probability, details are reported in Rule 1: Gaussian probability). The connections between the layers were not modified (i.e., the reference connectivity following Gaussian probability). The 2D reference for these 3D networks was a 2D SF network (Fig 1, $SF^{2D}$). The results of these configurations were compared against the ones obtained before with the reference connectivity for both 2D and 3D (Figs 3 and 1 ($G^{2D}$ and $G^{3D}$)).

Firstly, we evaluated the differences in the 2D fundamental units (Fig 5, *black boxes*). The introduction of the SF connectivity in the readout (bottom) layer induced an increment in the spiking and bursting rate. In particular, both the MFR and the MBR increased by + 38.30% (p = 0.03, Fig 5A) and by + 7.87% (p = 0.02, Fig 5B), respectively. The introduction of the SF topology (either in one or all the layers) in the 3D architecture (Fig 5, *green boxes*) produced a significant increase in the network's activity compared to the 3D reference ($G^{3D}$). In particular, neurons resulted to be more active (Fig 5A) by + 26.26% (p = 8.46 · 10⁻⁴) and by +51.91% (p = 7.55 · 10⁻⁷), in the $SF_{L0}^{3D}$ and $SF_{all}^{3D}$, respectively, with more frequent ($SF_{L0}^{3D}$: +10.42%, p = 3.14 · 10⁻⁸ and $SF_{all}^{3D}$: +12.74%, p = 7.88 · 10⁻⁴, Fig 5B) and longer ($SF_{L0}^{3D}$: +5.11%, p = 0.02 and $SF_{all}^{3D}$: +13.60%, p = 1.25 7.88 · 10⁻⁴, Fig 5C) bursts. In the $SF_{all}^{3D}$ configuration, even the network burst duration was affected, and displayed a significant increase of +6.37% (p = 0.006, Fig 5D). On the other hand, the $SF_{L0}^{3D}$ connectivity seemed to produce some effects on the $\tau_{rise}$, that displayed an increase of about + 20% (p = 0.02, Fig 5E).

Moving on to the direct comparison of all the activity features in the 2D and the 3D layouts, we found that only the $SF_{L0}^{3D}$ connectivity produced some changes in shape of the population events ($\tau_{rise}$, Fig 5E) and in the NBD (Fig 5D). It is worth noting that the alteration on the recruitment of the neurons ($\tau_{rise}$) was not observed *in vitro* [13]. Apart from these values in the sole $SF_{L0}^{3D}$, no other statistical difference, present between the 2D and the 3D architectures in the previous conditions, were highlighted. These results suggest that the SF topology has a

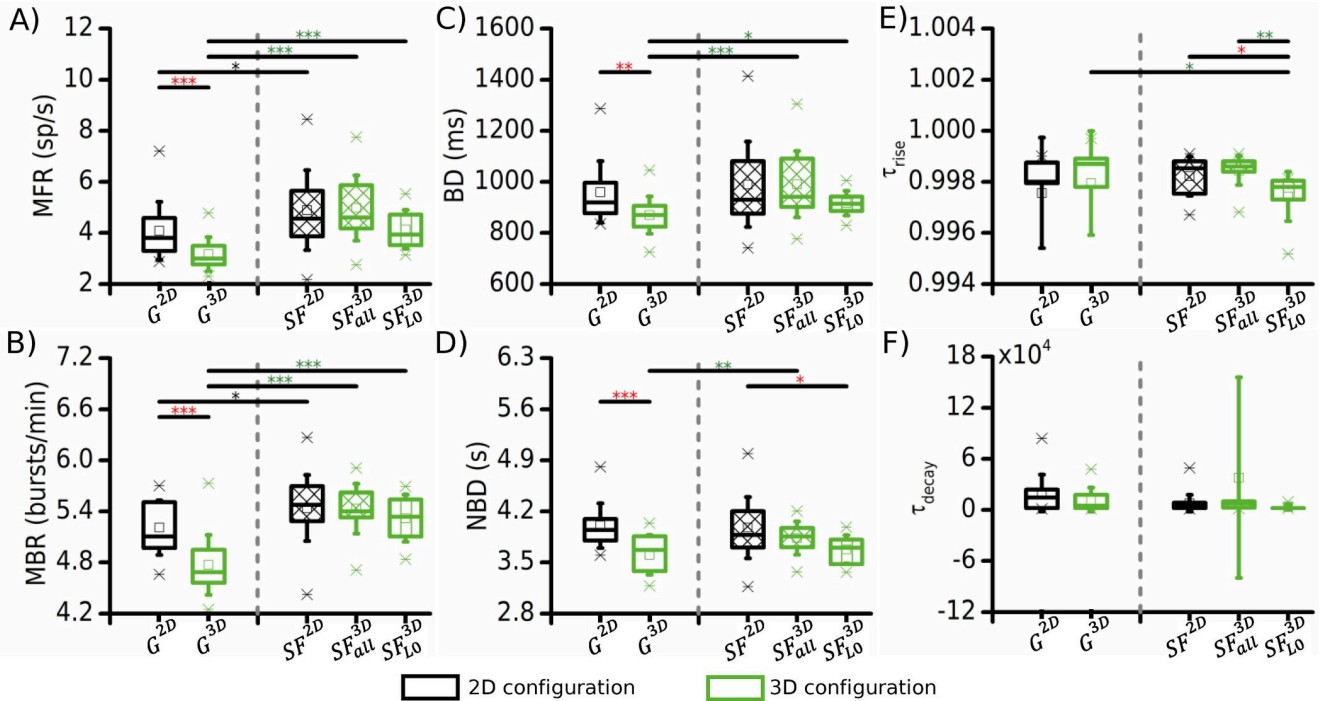

**Fig 5. Effect of the implemented single layer connectivity on the electrophysiological activity of 3D networks.** The 3D constructs (green boxes) with a SF connectivity either only in the readout layer ($SF_{LO}^{3D}$) or in every layer ($SF_{all}^{3D}$) were compared to the Gaussian ($G^{3D}$) reference topology. Similarly, the 2D controls (black) implemented a SF ($SF^{2D}$) and a Gaussian ($G^{2D}$) connectivity, respectively. (**A**) Mean firing rate, (**B**) mean bursting rate, (**C**) burst duration, (**D**) network burst duration, temporal time constants of the (**E**) rise and (**F**) decay phases of the network bursts. Black and green boxes stand for 2D and 3D networks, respectively. The asterisks are color coded to indicate the statistical differences between 2D and 3D networks (**red**), 2D networks with different connectivity (**black**), and 3D networks with different connectivity (**green**). (* refers to $0.01 < p < 0.05$, ** $0.001 < p < 0.01$, *** $p < 0.001$, Kruskal-Wallis non-parametric test).

prevalent effect on the dynamics of the network, and it mitigates the impact of the 3D architecture.

## Effect of the different implemented 3D connectivity rules

In this Section, we reported the results obtained in trying to tackle the issue of understanding the distribution of physical links along the *z*-direction in *in vitro* cultures when the only electrophysiological information available is from a planar readout. With a reverse engineering approach, we created different types of 3D connectivity (Sect 3D connectivity models in Materials and Methods), in particular by changing the position of the source neurons that generated the inter-layer connections. The selected configurations are: a Random (Fig 1, $G_{rnd}^{3D}$) one, where a random subset of neurons in each layer was selected to project to other layers; a Central (Fig 1, $G_C^{3D}$) one, where only a circular central cluster of neurons could connect with the other layers; and a Multiple Centers (Fig 1, $G_{5C}^{3D}$) one, where five different smaller projecting clusters were identified. We investigated such aspects of the 3D connectivity driven by the qualitative imaging characterization of *in vitro* 3D cultures where different possible scenarios of inter-layer connectivity emerged (S3 Fig, [8, 13]). The results were compared against the unrestricted reference approach that was employed in the previous analyses (Fig 1, $G^{3D}$), where each neuron projected afferences to the other layers and the control over the establishment of the connections themselves was a probabilistic one scaled over distance.

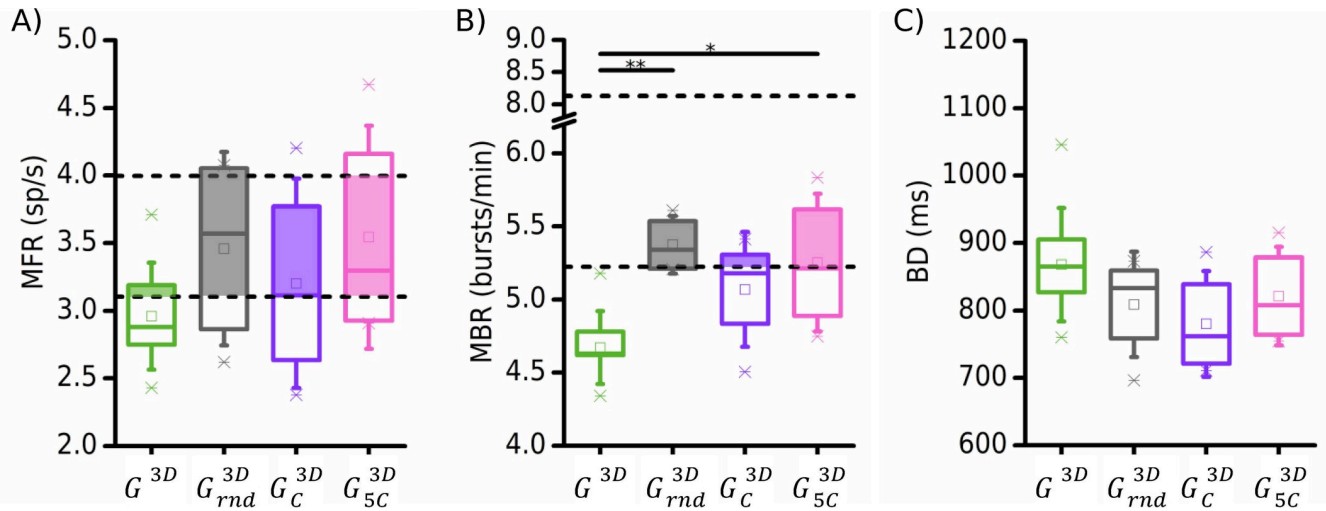

**Fig 6. Effect of the inter-layer connectivity on the exhibited dynamical features of the 3D networks. Four 3D topologies have been tested, namely unrestricted ($G^{3D}$), random ($G_{rnd}^{3D}$), and organized connectivity where source neurons are located in one single central area of the network ($G_C^{3D}$), or in multiple ones ($G_{5C}^{3D}$). The spatial organization of these configurations is depicted in Fig 7F–7H.** (A) Mean firing rate, (B) mean-bursting rate, and (C) burst duration of the different configurations. The box indicates the 25–75 percentile of the simulated data. The dashed black lines in the panel (A) and (B) indicate the 25–75 percentiles of the experimental data distributions. The colored shadow inside the box plots underlines the intersection between experiments and simulations. (* refers to 0.01<p<0.05, ** 0.001<p<0.01, Kruskal-Wallis non-parametric test).

We found that all the considered 3D connectivity rules produced only a mild increase (no statistical differences) in the firing rate with respect to the reference configuration ($G^{3D}$): the only appreciable difference was relative to a higher spread with respect to the control condition (Fig 6A). Concerning the bursting activity, we observed again a general increase of the bursting rate, that in the case of the random ($G_{rnd}^{3D}$) and with multiple centers source position ($G_{5C}^{3D}$) became significant: the MBR (Fig 6B) increased by +17.73% in the $G_{rnd}^{3D}$ configuration (p = 0.005, *grey boxes*) and by +13.91% in the $G_{5C}^{3D}$ one (p = 0.03, *pink boxes*). Eventually, the bursting events were slightly shorter in all the simulated configurations, however this decrease was not statistically significant (Fig 6C).

To try to find an answer to the main question of understanding what type of connectivity was more plausible established in a network leaning only on the planar readout, these results were compared against the data of 3D cortical *in vitro* cultures. Fig 6A and 6B show the 25–75 percentiles (dashed lines) of MFR and MBR values computed for *in vitro* cortical recordings. For both parameters, the simulated data overlapped with the recorded electrophysiological activity, especially in the $G_{rnd}^{3D}$ (grey) and $G_{5C}^{3D}$ (pink) configurations. The central distribution ($G_C^{3D}$, *purple*) lost some similarity when considering the MBR values. Instead, the $G^{3D}$ (green) connectivity did not fit well with the biological data, suggesting that the afferences to the other layers tend to follow a more structured connectivity rule, both on the number and on the position of the source cells.

## Discussion

In the present work, we examined the modulation of the electrophysiological activity of cortical networks brought by three-dimensional (3D) connectivity changes by means of an *in silico* model, taking as reference *in vitro* recordings of mature 3D assemblies coupled to Micro-Electrode Arrays (MEAs). Nowadays, 3D networks are the new frontier of *in vitro* models, allowing to overcome the intrinsic limitations of planar models in terms of altered morphology (e.g.,

flatten shape of the cell bodies [6]), microenvironment where neurons grow [32], as well as of stereotyped patterns of electrophysiological activity [8].

## Connectivity rules in simulated 3D networks: working hypotheses

With the proposed model, based on a network of synaptically connected leaky integrate-and-fire (LIF) neurons, we investigated the relevance (and organization) of the connectivity established first among the neurons belonging to the same layer and then between the layers. Indeed, connectivity is a property that is well recognized to trigger the dynamical states (spiking and bursting activity in the case under study) of the network in complex systems like the brain [33]. However, if from a phenomenological point of view the effect of a 3D connectivity in the emerging patterns of electrophysiological activity is clear, no quantitative speculations about the organization of the network have been made yet, with the sole exception of an experimental-computational paper by Bosi and coworkers. In that work, the authors reproduced calcium-imaging recordings in a 3D network of dissociated hippocampal neurons with a 3D *in silico* model characterized by long connections with higher values of clustering coefficient than the corresponding 2D controls [12].

In the present work, following a reverse engineering approach, we replicated some of the experimental features of 3D cortical recordings by designing different topological configurations inspired by functional and (partially) morphological findings. Indeed, by exploiting the rich repertoire of algorithms devised over the years for inferring functional connectivity [20, 34–38], it was assessed how 2D dissociated neuronal networks present features typical of complex networks. It has been claimed that hippocampal circuits show a preferential scale-free (SF) functional connectivity demonstrating the existence of hubs coordinating the activity of the entire assembly [31]. The network burst activation observed in dissociated cortical cultures is likely to be sustained by a SF organization of the connectivity of the network [39]. Always remaining in the functional domain, more complex topological configurations were found, like the arising of a modular connectivity [40] or rich-club topologies [29]. However, all these issues are limited to a 2D connectivity. In the present work, we implemented a neuronal network exploiting part of these functional experimental findings for the modelling of the bottom layer (the one coupled to the active area of MEAs in *in vitro* experiments) of our 3D neuronal population. Our results showed that the presence of a planar SF connectivity reduces the effects of the *z*-axis connections (Fig 5), suggesting that the presence of highly connected hub neurons is more relevant than the distribution of the cells in the 3D environment.

## Quantification of 3D

Another open question in the field of *in vitro* models is relative to what we can define "3D" or, in other words, at what height (and consequently at what degree of complexity of the experimental model) the switch from 2D to 3D can be appreciated. Our simulations foresee that at least 4 layers (about 200 μm) are necessary to observe a significant variation of the spiking and bursting statistics, and that the addition of further layers does not change the network dynamics (Fig 4). This result can be seen as a useful indication for the realization of experimental protocols. The Pautot method [7], based on the deposition of self-assembled layers of neurons and glass microbeads in a theoretically hexagonal structure, suffers the lack of homogeneity when the scaffold is too high (structural drawback). In perspective, our results suggest that fewer layers of the scaffolds than the ones previously used in the literature [8, 13] could be sufficient to generate diverse patterns of activity. If that were the case, a side but ethically important advantage would be the savings in terms of materials and cells to be used for each experimental session.

Moreover, although the dynamics of the network is modified by the more complex 3D organization, the recruitment and deactivation time of the collective behavior of the neuronal network are not affected (Fig 3). These results are in line with the considerations reported in [13], where it was proposed that the key parameter in the modulation of the network burst dynamics is the alteration of the E/I balance in the 3D connectivity due to different interacting populations (cortical and hippocampal in that study).

The last significant contribution of the model is relative to the organization of the connections in the 3D micro-environment. If insights came from functional investigations for the planar connectivity, the reconstruction of the 3D topology is completely blind. Still nowadays, the experimental investigations have not yet hypothesized possible connectivity patterns along the *z*-axis. In the present work, we simulated four kinds of 3D connectivity (Fig 6). A clear picture of a strong preferential connectivity rule did not emerge, as both the random and the more structured (central or multiple centers) topologies replicated the experimental spiking activity (Fig 6A) with a mild inclination towards the multiple centers when the bursting dynamics was considered (Fig 6B).

## Model limitations

Our model was tailored to the peculiar experimental configuration of 3D network organized in ordered and engineered layers obtained thanks to glass microbeads. These experimental constraints made the design of the model easier as it neglected the irregular 3D positioning of neurons assumed when they grow in amorphous scaffolds like hydrogels [41]. To simulate these 3D environments, more complex connectivity rules should be taken into account: the working hypothesis of an isotropic connectivity fails and as consequence, the topographic displacement of the neurons in the 3D space and of their connections requires the implementation of an anisotropic connectivity induced by the irregularity of the scaffold.

In addition, it is worth noting that the rationale behind the genesis of the computational model is based on a reductionist approach: we chose to describe the dynamics of the single neuron by means of a mono-compartment model whose electrophysiological properties are described by means of LIF equations. The choice to neglect a multi-compartment approach with complex morphology, as well as biophysical models of the ionic channels, is driven by the lack of experimental evidence relative to the *in vitro* biological substrate. As already stated, 3D networks from dissociated cultures establish a dense connectivity and we are not able to solve with sufficient precision where synapses are placed (i.e., in which segments) or how the dendritic arborization develops among the layers. Finally, during the dissociation procedure (cf. Sect, *In vitro* model), we do not have total control on where the neurons are extracted from (i.e., from which cortical layers) and this limits the choice of the neuronal morphology. Thus, although it could theoretically be possible to implement a 3D model with complex neuronal structures, the setting of its numerous parameters makes its applicability prohibitive for the goal of this work.

Despite the aforementioned limitations and tailored working hypotheses, the developed model is able to fit and mimic different features (MFR, MBR, percentage of inhibitory neurons) of 3D networks (and the relative differences). The lack of accuracy in the fitting of other statistics like the duration of the bursts and network bursts that result to be longer than the experimental ones, can partially be explained by the choice of the computational strategy (model) that does not provide any intrinsic description of the bursting activity in terms of ionic channels (e.g., persistent sodium channels) or by the absence of any form of facilitation/depression mechanisms (short term plasticity) at the level of the synaptic models.

## The use of functional connectivity to build *in silico* connectivity

From a connectivity point of view, the different structural configurations we implemented in the model are inspired by their relative functional counterparts. Indeed, this working hypothesis is based on the *in vivo* experimental findings that suggest the possibility to infer structural connections from the functional ones [21, 22] at the level of small assemblies of neurons. We are aware that this approach introduces some limits in the validity of our model. It is also worth mentioning that functional topological features are influenced by the detection algorithm, and the outcome is state-dependent. This last source of variability was clearly proved in *in vivo* sensory-motor tasks, where it was observed that the functional connectivity changes from resting state to task-induced conditions [42]. In *in vitro* assemblies, multiple modes of functional connectivity are much more limited because of the stereotyped patterns of activity that typically emerge from synaptic interactions [20] and the lack of external stimuli. A match between experimental and *in silico* structural connectivity could be achieved by modifying the experimental conditions of plating: a reduction of the cell density makes the identification of the network organization easier using microscopy techniques, and consequently could help the translation of the morphological observations to simulated connections.

## Model perspectives

In the present work, we developed a computational model of a 3D neuronal network driven by a peculiar experimental configuration achieved by stacking (ordered) layers of neurons [7]. As we wrote in the Introduction, this is not the only method used to build *in vitro* 3D networks, and more and more studies are exploiting the use of soft and amorphous materials as scaffold to allow the 3D neuritic growth [5]. Indeed, a change in the biological organization of the network requires variations in the *in silico* model too. However, the structure of the current model allows a relatively easy modification of parameters like the cell density and neuron position in the 3D space to accomplish the experimental requirements. On the other way round, the design of new connectivity patterns could be not trivial. An anisotropic scheme of connectivity will be necessary to map the inhomogeneity of the space where neurons live [43].

In perspective, the proposed *in silico* model could also be exploited to investigate other open questions related to 3D ensembles and can be customized for peculiar experimental setup. To prove such potentiality, we performed preliminary simulations of an experimental condition where (at least) two 3D neuronal ensembles are interconnected by means of bundles of links originating from the bottom layers (S4A Fig). In this way, in addition to the 3D connectivity, also the modularity topological feature was considered [44]. This configuration (also reproduced in *in vitro* experimental models) better mimics the large-scale network organization of mammalian brain and allows to reproduce a wider repertoire of dynamics [45]. Preliminary simulations show that from the similarity maps (S4B Fig), we can observe that whilst one of the modules continues showing the highest similarity values within the module, the network burst initiation events of the second module present lower inner similarity values than the ones obtained with the first module suggesting that the propagation of the signal is greatly disrupted by the addition of modularity.

Finally, the last ingredients that the model should take into account is relative to the heterogeneity of 3D interconnected neuronal networks. In 2020, Brofiga and co-workers developed an experimental model where cortical and hippocampal neurons were chronically coupled to MEAs defining interconnected 3D heterogeneous sub-populations [13]. To explain the contribution of connectivity and neuronal heterogeneity, the authors speculated that hippocampal neurons projected strong inhibitory links to the cortical ensemble and observed a significant reduction in the duration of the network burst, but they were not able to untangle the role of

each of the two components (three-dimensionality and heterogeneity). With the aim of developing brain-on-a-chip models where neurons extracted from different brain areas (e.g., cortical-hippocampal assemblies) interact in a 3D fashion, our model could help to understand how the connections between the neuronal populations are physically and functionally organized and which are their targets.

## Materials and methods

### Ethics statement

The experimental protocol for *in vitro* cultures was approved by the European Animal Care Legislation (2010/63/EU), by the Italian Ministry of Health in accordance with the D.L. 116/1992 and by the guidelines of the University of Genova (Prot. 75F11.N.6JI, 08/08/18).

### Computational model

The model aims at generating patterns of electrophysiological activity that can be related and compared to those emerging from *in vitro* 2D and 3D neuronal populations coupled to planar MEAs and at maintaining a comparable spatial distribution. To implement these aspects, the model was subdivided into the following problems: (i) creation of a fundamental 2D unit and (ii) the implementation of the 3D layout.

### Single neuron model and 2D network

The dynamics of each cell was modeled by means of leaky integrate-and-fire (LIF) neurons (Eq (1)):

$$\tau_m \frac{dV_m}{dt} = (E_L - Vm) + \frac{1}{g_L}\left(I_{noise} + I_{syn}\right) \tag{1}$$

where $V_m$ is the membrane potential, $I_{noise}$ models the afferences to the neurons, $I_{syn}$ is the synaptic current (cf. Sect Synaptic Models), $g_L$ the leakage conductance, $E_L$ the resting potential, and $\tau_m$ the time constant of the neuron. The numerical values of the parameters of the neuron model are listed in Table 1.

The neurons were synaptically connected in order to investigate the effects produced by the 3D connectivity on the activity of the network. The relative simplicity of the neuronal model allowed focusing only on the impact of the complex network organization. The fundamental network (Fig 7A) was composed of 1'091 cells disposed on a square grid with area of approximatively 0.7 mm$^2$, giving a cell density of 1'480 cell/mm$^2$, which is comparable to the value obtained for *in vitro* cortical cells of about 1'500 cell/mm$^2$ (cf. Sect *In vitro* model). Neurons were formally divided into excitatory (Exc) and inhibitory (Inh) cells based on their post-synaptic effect (cf. Sect Synaptic models). The percentage of inhibitory cells in the network was set to 20% (S1 Fig) to mirror *in vitro* dynamics (cf. Results, [26]). Inhibitory cells were randomly placed in the 2D grid (Fig 7A, *blue dots*) among the excitatory cells (Fig 7A, *red dots*).

The position of each neuron on the grid-like layout was slightly modified by adding some noise on the *x* and *y* parameters, specific to each neuron, to mimic the random distribution of cells in *in vitro* cultures. For the whole population, the axonal velocity and the refractory period were set to 0.5 mm/ms and 5 ms, respectively [46, 47]. The physiological afferences were modeled with a current $I_{noise}$, coupled with a leakage conductance $g_L$. The noisy input values were varied to find the optimal configuration to trigger subthreshold fluctuations, necessary to ensure the onset of action potentials and thus the spontaneous activity of the network. The $I_{noise}$ value was chosen randomly every millisecond for each neuron (Table 1).

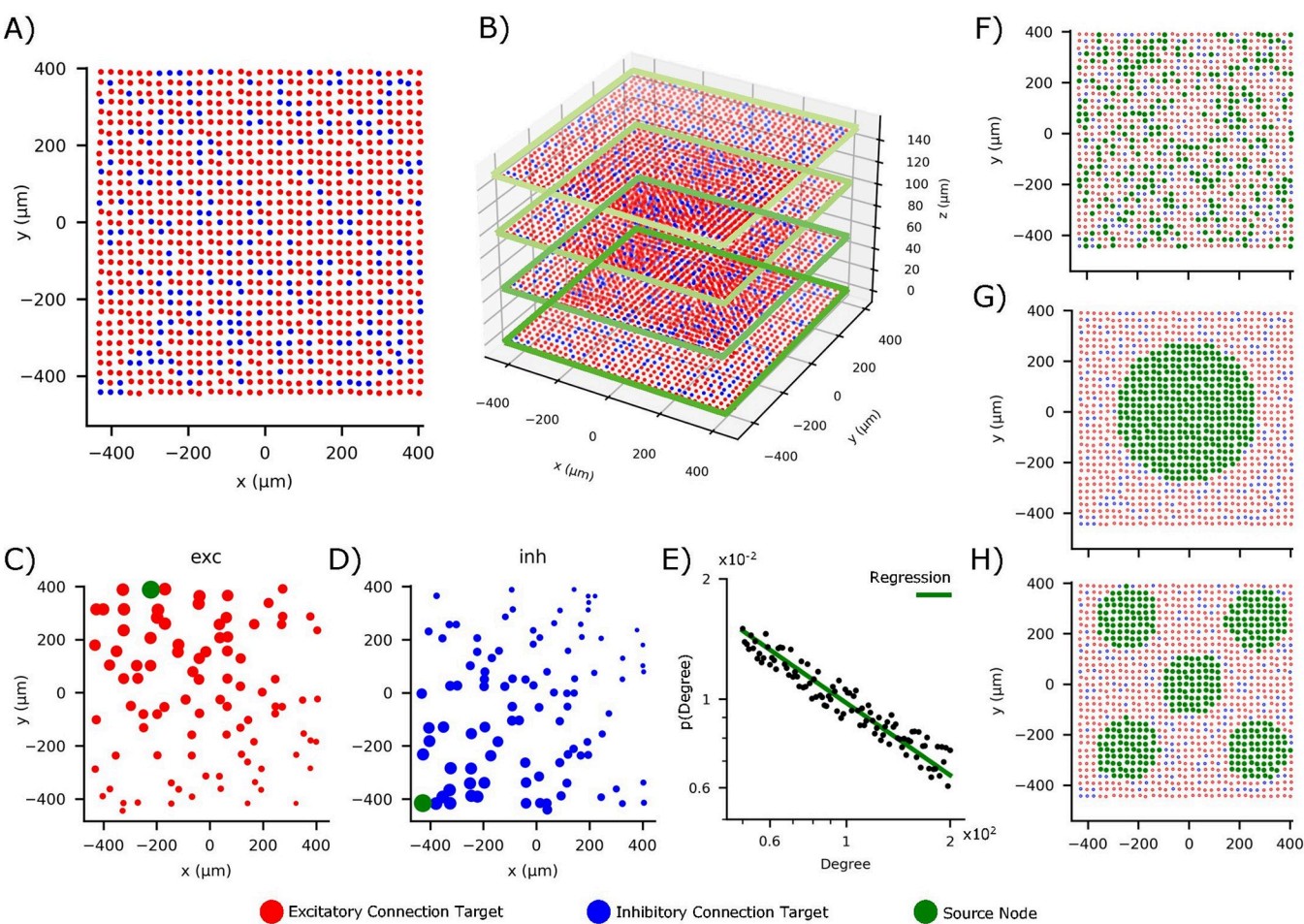

**Fig 7. Topological properties of the simulated neuronal networks.** (**A**) Neurons position in a 2D layer. Excitatory (**red**) and inhibitory (**blue**) neurons are randomly arranged on a surface of 0.7 mm², defining the fundamental unit of the network. (**B**) Dense and compact 3D networks obtained by piling up different layers (only 4 in this sketch, for the sake of clarity) organized as in (**A**) on the 3D space. (**C-D**) Spatial distributions of the synaptic connections for excitatory (**red**) and inhibitory (**blue**) links in each layer. Considering a source neuron labeled in green, it projects connections whose efficacy (size of the colored target neurons) decreases as a function of the distance in according to Eq (3). (**E**) Degree distribution of the scale-free network showing a power-law with characteristic exponent equal to -0.61, in accordance with [19]. The distribution was implemented either only in the readout layer, in all the layers or in none of them. The connectivity among the layers was modeled according to different possible configurations depicted in the panels (**F-H**). In particular (**F**) shows a random positioning of the source neurons that project synaptic connections to the other (up or down) layers ($G_{rnd}^{3D}$). In (**G-H**), the source neurons for the inter-layer connections are arranged in a single central area of the plane (**G**, $G_C^{3D}$) or distributed among five smaller confined areas (**H**, $G_{5C}^{3D}$). The number of links is kept constant in the three configurations to allow significant comparisons.

## Synaptic model

The synaptic effect induced by excitatory and inhibitory synapses was modelled with two different exponentially decreasing currents $I_{synEX}$ and $I_{synIN}$, whose dynamics are described by Eqs (2A) and (2B), respectively:

$$\frac{dI_{synEX}}{dt} = -\frac{I_{synEX}}{\tau} \tag{2A}$$

$$\frac{dI_{synIN}}{dt} = -\frac{I_{synIN}}{\tau} \tag{2B}$$

where $\tau$ is the time constant of the synapses, set at 10 ms. When a pre-synaptic action potential

reached the post-synaptic neuron, the post-synaptic current $I_{synEX}$ ($I_{synIN}$) was incremented (decremented) by a fixed value $\alpha_{EX}$ ($\alpha_{IN}$) multiplied by a distance-dependent synaptic weight $w$. The last was computed as in Eq (3):

$$w = c \cdot \frac{1}{\sqrt{2\pi\sigma^2}} \cdot \exp\left(-\frac{(x_{pre} - x_{post})^2 + (y_{pre} - y_{post})^2}{2\sigma^2}\right) \tag{3}$$

where $\sigma$ is the spatial standard deviation (set at 1000 μm) and $c$ (set at $\frac{1000\ mV}{g_L}$) is a correcting factor to adapt the amplitude and the unit of the weight to the current values. The values $\alpha_{EX}$ ($\alpha_{IN}$) were set at 12 and 16, respectively (S5 Fig). The response of the post-synaptic neuron to an incoming action potential was delayed and weakened accordingly to the distance of the source cell and the axonal velocity.

## 2D Connectivity models

Different connectivity rules (inspired by the experimental findings on functional connectivity) were applied and studied within the fundamental 2D unit. In any of the following cases, autapses were avoided as they are unlikely to form in dissociated neuronal cultures [48]. Only excitatory to excitatory, excitatory to inhibitory, and inhibitory to excitatory connections were implemented in the present work driven by experimental findings.

At the basis of all the connectivity implementations, a probabilistic connectivity rule was created as in the following. The probability to establish a connection between two neurons was modulated according to a Gaussian decay over distance:

$$p = p_{MAX} \cdot \exp\left(-\frac{(x_{pre} - x_{post})^2 + (y_{pre} - y_{post})^2}{2\sigma^2}\right) \tag{4}$$

where $p_{MAX}$ is the maximum probability for each connection. We swept $p_{MAX}$ to find the optimal value that gives rise to a good network activity that mirrors the one found *in vitro* (Table 1).

### Rule 1: Gaussian (*G*) connectivity

An additional condition was added to the previous rule that restricts the "physical" links between neurons. The number of outgoing connections ($N_{outgoing}$) derived by Eq (4) was evaluated for each neuron and compared to a threshold set at 8% of the total number of cells. If $N_{outgoing}$ exceeded the threshold by a value $\beta$, $\beta$ connections were randomly deleted from the weight matrix. The connections established from exemplary source excitatory and inhibitory nodes (green dots) are depicted in Fig 7C and 7D, respectively. The size of the target neurons (red for excitatory connections, blue for inhibitory ones) represents the connection weight (Eq (3)).

### Rule 2: Scale-free (*SF*) connectivity

To implement a scale-free (SF) connectivity organization, the probabilistic rule of Eq (4) was integrated with a control over the degree established for each neuron. A custom function was created to fit the one of the typical degree distributions found in *in vitro* cortical cultures [19]. The function required the setting of four parameters: (i) the slope $\alpha$ of the fitting of the degree distribution, set at -0.61; (ii) the upper and (iii) lower limits of the distribution, set at 50 and 200, respectively; and (iv) the number of degree values to extract, set at 100. The function extracted the probabilities of each degree value and, from these, it computed the threshold of outgoing connections, $N_{outgoing}^{max}$, for each neuron (Fig 7E). This threshold was then compared to

the number of established connections, which were randomly deleted from the weight matrix if they exceed $N_{outgoing}^{max}$. To reproduce what happens experimentally, the position of the high-degree neurons (hub neurons) in the 2D layout was random [29].

### 3D connectivity models

The 3D configuration was generated by piling up several 2D units as described in the first Section (Fig 7B), inspired by the experimental procedure of [7]. A $z$ parameter was added to the intrinsic parameters of the cells to set the height of each layer. The distance between different layers was set to 50 μm to mimic *in vitro* scaffolds made up of glass microbeads [8, 13]. These parameters yield to a final density of about $30/000 \frac{cell}{mm^3}$. To evaluate the role of the total height and the numbers of involved layers, networks with 2 up to 6 layers were simulated.

Generally, a connectivity like the one of the 2D fundamental unit (cf. Sect Connectivity models) was implemented also in the 3D case. A probabilistic control over distance (which in this case included the $z$-coordinate) was applied to evaluate the connection between two neurons pertaining to different layers. The maximum probability of connections was in this case set to $\frac{1}{3}$ of the maximum probability of connection $p_{MAX}$ of the 2D fundamental unit. Moreover, much as in the 2D case, we inserted a restriction over the number of outgoing connections.

Assuming these hypotheses, we evaluated different dispositions of the neurons that generated the 3D connections. The underlying issue for this change was that it is not yet experimentally understood how the physical links are distributed in the $z$-direction in *in vitro* cultures. Therefore, the first, and reference, approach was random and unrestricted ($G^{3D}$). Then, we explored the effect of limiting the number of neurons that generated the 3D links ($G_{rnd}^{3D}$, Fig 7F). Afterwards, we investigated whether the connections were originated in specific regions of the network. Two different types of clusters were examined: in the first case (Fig 7G), the inter-layer connections were originated from neurons positioned in a 278-μm radius from the center of the culture itself ($G_C^{3D}$); in the second (Fig 7H), the source neurons were located in 129-μm radius circles centered in 5 different locations of the culture ($G_{5C}^{3D}$). In all the new configurations, the number of source neurons for inter-layers connections results to be around 360, that is 33% of the total population of each layer.

As for the experimental counterpart, we considered only the *in silico* data coming from the bottom (or readout) layer for the analyses unless otherwise stated.

### Dataset

The presented results come from a dataset made up of $n$ = 20 2D networks with Rule 1 connectivity ($G^{2D}$), $n$ = 28 2D networks with Rule 2 connectivity ($SF^{2D}$), $n$ = 25 3D networks ($G^{3D}$) equally divided among the considered number of layers (2 to 6), $n$ = 22 3D networks with Rule 2 connectivity ($SF_{all}^{3D}$) in all the layers equally divided among the considered number of layers (2 to 6), $n$ = 12 3D networks with Rule 2 connectivity in the readout layer and Rule 1 in the other layers ($SF_{L0}^{3D}$), and $n$ = 12 four-layer 3D networks equally divided among random ($G_{rnd}^{3D}$), central ($G_C^{3D}$), and multiple centers ($G_{5C}^{3D}$) connectivity. The *in vitro* recordings used to tailor the model parameters came from $n$ = 5 3D cortical cultures in their mature stage of development (18 days *in vitro*).

### Simulation environment

The neuronal network models were created using the Python programming language and exploiting the Brian simulator package [49], a tool specifically designed for creating spiking

neuronal networks. All ordinary differential equations were solved using the exponential Euler method, with a time step of 0.1 ms. Spikes were recorded with sampling frequency of 10 kHz, in accordance with the experimental recordings. All results presented were taken from simulations producing 60 seconds of data, recorded after 10 seconds of simulation necessary for the network to settle. The parameters used for the simulations were either extrapolated from the literature or calibrated with empirical trials, where the values of the parameters were chosen in order to guarantee a stable and physiological dynamics of the cells. A list of the swept parameters can be found in the supplementary information (S5 Fig and S1 Table).

## Data analysis

The simulated electrophysiological activity was analyzed using algorithms developed in Matlab (The Mathworks, Natik, US). From the spike trains of both the excitatory and inhibitory populations of the readout layer, burst events were identified by means of the string method devised in [50] by setting to 5 the minimum number of spikes needed to be classified as a burst and to 100 ms the maximum inter-spike interval into a burst. To quantitatively characterize the spiking and bursting activity, the following parameters were computed: (i) the mean firing rate (MFR) and (ii) mean bursting rate (MBR), that are the mean number of spikes/bursts per second/minute averaged over the number of active neurons; (iii) the burst duration (BD), which is the temporal length of a burst. A neuron was considered active if it generated at least one spike in 10 s (MFR $> 0.1 \frac{sp}{s}$) and at least four bursts in 1 min (MBR $> 4 \frac{bursts}{min}$).

The choral activity of the whole population was evaluated by analyzing the network bursts, identified with the algorithm described in [51]. The algorithm requires two thresholds for the detection of these population events: (i) the maximum interval that occurs between two consecutive events (set at 100 ms) and (ii) minimum number of involved nodes (set at 20% of the total number of nodes). From the network burst train, we computed the network burst duration (NBD), i.e., the temporal duration of the population events. Information about the shape of the network bursts was computed from the average spike time histograms (STH). The instantaneous firing rate (IFR) of each network burst were aligned by evaluating the delay that produces the maximum absolute value of cross-correlation between a randomly chosen IFR trace and the remaining network bursts and temporally shifting the data accordingly. The average STH was accepted when the correlation coefficient was higher than 0.4 [52]. Then, the rising and decaying phases were individually fitted to evaluate the temporal evolution with the following exponential functions, respectively:

$$r(t) = a_0 e^{(b_0 - 1)t} + a_1 e^{(b_1 - 1)t} + a_3 \tag{5}$$

$$s(t) = c_0 e^{d_0 t} + c_1 e^{d_1 t} \tag{6}$$

The smaller of the two exponents $b_0$ and $b_1$ of Eq (5) defines the neuron recruitment rate [52], while the smaller between $d_0$ and $d_1$ of Eq (6) describes the network modulation during the population events (modification of [26]).

## Statistical analysis

Statistical analysis was performed using Origin (Origin Lab Northampton, Ma) with the non-parametric Kruskal–Wallis test, as data do not follow a normal distribution (evaluated by the Kolmogorov-Smirnov normality test). Significance levels were set at $p < 0.05$. The box plots representation indicates the 25–75 percentile (box), the standard deviation (whiskers), the mean (square), and the median (line) values. For the last section, to make the data comparable,

different subsets of the 3D reference configurations with dimensions comparable to the other data were tested, each resulting statistically different.

## In vitro model

Cortical tissue was obtained from Sprague-Dawley rat embryos at gestational day 18 (E18). Single cells were obtained by enzymatic dissociation followed by a mechanical one. The enzymatic solution consisted in Hank's solution with 0.125% Trypsin and 0.05% DNAase and was activated for 20 minutes at 37˚C. This process was quenched by adding culture medium supplemented with 10% fetal bovine serum, followed by a mechanical trituration with fire-polished Pasteur pipette. The cells were plated on pre-coated planar MEAs to extracellularly record the electrophysiological activity of the networks [19]. To create the 3D structure, glass microbeads (40 μm in nominal diameter) were plated over the MEA as described in [9]. Briefly, the pre-coated beads were moved to a multiwell plates to form a uniform layer for cell seeding. Microbeads self-assembled in a hexagonal geometrical structure, leaving only two thirds of the surface of the beads exposed to the cells, which will not be as regularly arranged in the plane as the beads. They were plated on the beads to obtain a final nominal density of 1'500 cell/mm$^2$. After 6 hours, the beads and attached cells were moved onto the cell monolayer in the MEAs, to form a four/five-layer final structure. Half of the medium was replaced every week. Cell cultures were maintained in a humidified incubator at 37˚C supplied with 5% $CO_2$ for about 3 weeks, until networks maturation.

## Supporting information

**S1 Fig. A physiological balance between excitation and inhibition guarantees *in vitro*-like patterns of electrophysiological activity where bursting and spiking activity coexist.** 60-s simulated spontaneous electrophysiological activity of representative 2D networks as a function of the percentage of inhibitory neurons. (**A**) 50%, (**B**) 40%, (**C**) 30%, (**D**) 10%, and (**E**) 20% of inhibitory neurons. This last configuration allows to reproduce values of firing and bursting rates comparable with the experimental recordings of mature 2D cortical cultures as well as the presence of population events (network bursts) involving most of the neurons of the network.
(TIF)

**S2 Fig. Involvement of the network in population events.** (**A**) Number of involved electrodes in the network burst events in 2D (**black**) and 3D (**green**) networks (*** refers to p<0.001, Kruskal-Wallis non-parametric test). (**B-D**) Color-coded normalized similarity maps, evaluated with the Victor Purpura distance, within 3 exemplary of 4-layer 3D networks. The normalization was done on the maximum similarity. From the maps, the three higher values of similarity were extracted and classified based on the step distance between two layers, (e.g., two consecutive layers are considered as 1-step distant; layer 1 and layer 4 are considered 3-step apart). Three different propagation modes emerged, each represented with an exemplary normalized similarity map and a relative pie chart, which indicates the percentage of times the higher similarity values occur between consecutive layers (d1), layers 2-step apart (d2), and layers 3-step apart (d3). Each identified propagation mode was exhibited by n = 3 simulated 4-layer networks ($G^{3D}$).
(TIF)

**S3 Fig. 3D *in vitro* culture.** Mature 3D culture (18 DIV) on glass microbeads, stained with anti-NeuN antibody to label neuronal nuclear protein.
(TIF)

**S4 Fig. Similarity of modular cultures.** (**A**) representation of the disposition of the neurons in the network. Two different populations (green and yellow) were created as with the Gaussian connectivity ($G^{2D}$ and $G^{3D}$). The two modules were connected on the readout layer level to mimic the experimental model devised in [13], where two different 3D populations were interconnected by microchannels. To imitate the physical constraints, the populations were 250 μm apart and the gaussian function that regulates the synaptic weight *w* and the probability of connection *p* was modified to introduce anisotropy. In particular, two different σ were implemented, a transversal one that is a multiple of the distance between two channels (50 μm) and a longitudinal one, that adds to that the distance between the two modules. (**B**) Color-coded similarity maps, evaluated with the Victor Purpura distance (S1 File), within the two modules (green on the left, yellow on the right). The arrow and value on the top indicate the similarity value between the readout layers of the two modules.
(TIF)

**S5 Fig. Connectivity parameters tuning.** Effect of the mean synaptic inhibitory weights on the (**A**) mean firing rate (MFR) and (**B**) mean bursting rate (MBR). The effect of the percentage of pruning synaptic connections was evaluated and its effect evaluated on the (**C**) MFR and (**D**) MBR. Red, blue and black colors identify the results relative to the excitatory, inhibitory, and the total neurons, respectively.
(TIF)

**S1 Table. List of parameters swept for tuning.**
(TIF)

**S1 File. Evaluation of similarity method.** Method for the extraction and evaluation of the similarity maps.
(DOCX)

## Acknowledgments

The authors are grateful to Dr Leonardo Vernazza for running some simulations in the setting up of the model parameters. The authors wish to thank Dr. Mariateresa Tedesco for the excellent cells.

## Author Contributions

**Conceptualization:** Paolo Massobrio.

**Data curation:** Martina Brofiga.

**Formal analysis:** Martina Brofiga.

**Investigation:** Francesca Callegari, Paolo Massobrio.

**Methodology:** Martina Brofiga.

**Project administration:** Paolo Massobrio.

**Resources:** Paolo Massobrio.

**Software:** Francesca Callegari.

**Supervision:** Paolo Massobrio.

**Validation:** Francesca Callegari, Martina Brofiga.

**Visualization:** Martina Brofiga.

**Writing – original draft:** Francesca Callegari, Martina Brofiga, Paolo Massobrio.

**Writing – review & editing:** Francesca Callegari, Paolo Massobrio.

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
