## [Decision Letter · Decision Letter 0]

13 Sep 2022

Dear Prof. Massobrio,

Thank you very much for submitting your manuscript "Modeling the three-dimensional connectivity of in vitro cortical ensembles coupled to Micro-Electrode Arrays" for consideration at PLOS Computational Biology.

As with all papers reviewed by the journal, your manuscript was reviewed by members of the editorial board and by several independent reviewers. In light of the reviews (below this email), we would like to invite the resubmission of a significantly-revised version that takes into account the reviewers' comments.

Please make also sure that the code necessary to reproduce the results in the paper is freely available in a third party repository.

We cannot make any decision about publication until we have seen the revised manuscript and your response to the reviewers' comments. Your revised manuscript is also likely to be sent to reviewers for further evaluation.

Sincerely,

Daniele Marinazzo

Section Editor

PLOS Computational Biology

Reviewer's Responses to Questions

**Comments to the Authors:**

Reviewer #1: This is an interesting article, well and written, and deserves publication in PLoS Computational Biology. However, I have some concerns that must be addressed before publication.

1. The introduction is overall clear and introduces the problem well. However, authors could mention (either in the introduction or later in the discussion) the efforts made to understand the relation between structural and functional connectivity beyond the general reference 21. For instance, Ludl et al. (Front Comput Neurosci 2020) showed using numerical simulations that strong spatial constraints help dictating network dynamics and functional traits. Additionally, my impression (and I think consensus from the community) is that the relation between structure and function is not direct given the intrinsic nonlinear behavior of neuronal systems and the ubiquitous presence of noise. For instance, using calcium imaging, Yamamoto et al. (Sci Adv 2018) observed that networks designed structurally in a modular way could show very different dynamics and functionality depending on the coupling between the modules, suggesting that small changes in structural connectivity could render very different functional networks. Somehow these aspects have to be reflected in the manuscript. Otherwise, one gets the impression that structural connectivity can be predicted from just analyzing the dynamics.

2. In Methods, the authors explain that they design 2D networks that are vertically coupled to shape the final 3D system. If I understand correctly, structural connectivity within each 2D layer is imposed to be scale free based on Ref. [18]. I found this worrisome, since Ref. [18] provides functional data for connectivity distributions, not structural. So somehow the authors are assuming that functional connectivity is a good representation of the structural one, which I think is not correct. At least it should be explained that a strong assumption is made. It is also difficult for me to understand why identical neurons that homogeneously cover a 2D area shape a scale free structural network without any kind of connectivity guidance or constraints.

3. Related to this, I note for instance that Orlandi et al (Nat. Phys 2013) showed numerically that a Gaussian-like structural connectivity reflects well the dynamic behavior of 2D experimental cultures. In their simulations they observed that scale free behavior emerged in the dynamics of the network due to the interplay of connectivity, metric correlations and noise. Thus, I think that in the present article, the authors should test what happens when they just assume that structural connectivity in each 2D layer is Gaussian or just dictated by the distance-dependent connectivity rule, without imposing a scale free connectivity. My feeling is that the results will not change much, strengthening the importance of the interlayer connectivity.

4. Most of the panels in Fig. 1 are not necessary. There many works in the literature describing how network bursting increases as the inhibitory-to-excitatory ratio is reduced. What I missed in Fig. 1 are raster plots of the full 3D system comparing the interlayer connectivity schemes. I would expect to see that the different layers do not necessarily activate synchronously, which I think is an interesting message. The authors could then mark which of the layers in the 3D network raster plot correspond to the “MEA readout” layer that they wish to compare with experiments.

5. I got confused in Fig. 4. The authors seem to suddenly show experimental data (since they mention in Methods that they have experimental data), but the discussion of the figure seems to be about simulations. I got lost. Can the authors put both experimental and numerical data in that panel? Additionally, in line 234-235, the sentence “As a tight correspondence between functional and structural connectivity was observed…” is very strong sentence if it refers to experimental data. Where is the proof that structural connectivity and functional one coincide?

6. The discussion is nice, but the authors could extend it a little bit by commenting the limitations and extensions of their work. For instance, they could discuss on the possibility to break the synchronous bursting behavior by adding modularity in the 2D layers and by altering the range of interlayer connections, so some neurons can connect beyond their immediate neighborhood, somehow approaching the cortical columns observed in the brain. I have the feeling that the 3D system they obtain can be viewed as a 2D ‘readout’ network with effective bombardment of activity from the other 2D networks, which disrupts its collective behavior.

Other minor aspects.

1. Details are necessary in the description of the box plots of the figures. How many data points are in the distributions? Is the data coming from a single numerical repetition or several ones? Are results consistent among repetitions?

2. In the caption of Fig. 7, it should be clarified that the distribution of panel E refers to a single 2D layer, not the entire 3D construction.

3. The caption of Fig. 2 should clarify that the 3D data is actually the 2D readout layer, not the whole 3D population.

4. Line 416: with characteristics exponent -> with characteristic exponent.

5. It should be clearer in Fig. 4 that the shown data is ‘in vitro’. Maybe in the title, replace ‘cortical assemblies’ by ‘cortical assemblies in vitro’.

Reviewer #2: The study of Callegari and colleagues is a 3D neuronal network model trying to simulate and reproduce the experimental conditions/data from Pautot and Frega papers. This is a very interesting topic, since 3D in-vitro neuronal networks are key experimental models overcoming previous 2D models (classical dissociated neuronal cultures) to study brain functioning and diseases. In-vitro 3D networks show broader repertoire of dynamics compared to 2D networks possibly better reproducing brain functioning mechanisms. 3D networks from dissociated neurons or brain organoids are the new frontiers of in-vitro brain models. 3D networks models reproducing 3D cultures are still lacking, therefore Callegari et all focus and explore a high interest topic both for computational and experimental neuroscientists. The main result is that the introduction of three-dimensionality induces a decrease in both firing and bursting rates with respect to 2D models. In addition, the authors found that at least four layers are necessary to induce a change in the dynamical state of the network.

My first major comment is that the there is no mention in the model and in the experimental data on the rich dynamical regimes observed in the 3D case. I mean 3D are better than 2D networks because dynamics are different and not only in terms of single neuron and network firing and burst rates, but also in term of terms of neural synchronizations. The 3D network display dynamics which go beyond the stereotyped 2D networks activity characterized by network bursts which recruit the overall circuit. This is a key aspect which should be also addressed in the work.

My other main major comment is that in the actual state, the paper is extremely difficult to be read and understood, and it requires a very challenging ping-pong between the methods, the main studies to which the work refer to (Frega et all, Pautot et all), and the data shown in fig 4 (are the results from another paper or generated and freshly within this research?).

Also the introduction and the paragraph on the scale-free connectivity is very hard to be understood. What are the four conditions considered?

Same story when the scale-free 2D networks results are compared to SFALL and SF0.

The high mix of data and models with lack of proper explanation undermine the richness of the available experimental data (2D and 3D).

So a major reshape of the MS is definitely needed. The model and its configurations can not be only presented in the methods section. More details are needed within the MS especially in the opening part of the results, where all that is written now is confusing rather than helping to understand what is later presented.

More detailed comments follow, but a drastic reshape of the MS is needed to explain exactly how the model is built and which configurations are tested and to which experimental data you refer to.

ABSTRACT

“In addition, we found that at least four layers are necessary to induce a change in the dynamical state of the network and that a scale-free connectivity within a layer tends to

reduce the effects of the 3D organization.” this sentence is disturbing in the abstract, make it not clear in a way. Also, this refers only to the bottom layer as scale free? so it is not a general within layer organization.

INTRO

Line 68 to 74

Self-citations are fine when also a broad view is presented. Most likely the experimental model you mentioned, and your following work (Frega et al) were the first works on 3D cultures and recording of their dynamics. However, I find this self-citation too strong in terms of relevance in the beginning of the intro. I suggest to smooth it or narrow to this point later.

Indeed, you have many more ways to create 3D networks from dissociated neurons, so you should more broadly introduce them also in relation to their dynamic regimes. do 3D networks from dissociated neurons exhibit the same dynamical regime you try to reproduce? address this in intro to make your study of more broad interest. Otherwise, this narrows too much the interest to the Frega et all experimental model, limiting the interest of your work to only that specific 3D model. i think the study here (https://www.ncbi.nlm.nih.gov/pmc/articles/PMC5220075/) is an example of how 3D network regimes are different from 2D in line with the Frega paper results. Also the mentioned study uses a non constrained spatial configuration as the one imposed by the beads and also the mentioned study is based on optical methodolgy which offers single cell resolution, so makes the overall view non limited to electrical recordings.

Please broaden your intro to make it of more general interest.

Line 84-92

maybe discuss other electrophysiological methods to record 3D cultures. as I suggested before calcium imaging and optical measurement should be in principle able to record at different depth, with single cell resolution, overcoming intrinsic planar electrode array limitations (lack of information in terms of recording of cell location and type).

In general the work you present is too narrowed to the papers of Frega, Bosi and Pautot. These are just starting points and you want to offer broader message indeed.

Line 102-104

explain better, not clear, what you mean average length? spatial? mean path length is topological, not spatial. so better in case you clarify which length you refer to and make it clear. clustering, is spatial clustering or topological clustering? Please make this more clear just to better interpret the model, since you spend some sentences over it and you make clear focus on it

line 110-113

not clear that you refer to your present work

line 117-119

which in vitro models, 3d or 2d? the last part of the introduction summarizing the work is not clear at all. I mean obviously after you read the paper it becomes better understandable, but as an intro which should facilitate to approach the overall work, surely this is not clear

RESULTS

At the beginning of the results, you have to better explain, as in the introduction, that you use as fundamental unit your model of 2D networks (single layer) and you next interconnect layers.

Also the bottom layer is spatially unconstrained so it is the only layer that you also model it as a scale-free network (apart the case of SFall). All other layers are added next. Interconnectivity between layers is free or spatially constrained to neurons located in specific areas (two conditions are modelled). All this has to be clear at the beginning of the results, since this is the core of your model, so most of this part can be moved from methods to results section indeed.

Line 125-126: “Firstly, we examined the intrinsic dynamics of the generated 126 model and tested the effect of the 3D configuration on the activity of the network.” What do you mean here? This is totally unclear

Line 130-134: “Then, the role of the established connectivity within each layer was explored, to test whether the functional characteristics observed in MEA recordings underlie an effective connectivity also in the 3D configuration. A similar analysis was carried out on the inter-layer connections with the rationale of inferring the in vitro connectivity from the activity patterns recorded at the readout layer. Eventually, the data of the different configurations were compared to in vitro data.” This is not understandable to a person that did not read the paper. This means that is useless in the present way. So as I suggested the first part of the results has to explain the model and the steps you will go over in the results.

Figure 1: The only clear raster plot of fig 1 is the panel E. The rest of the rasters are not really visible. Indeed panel D is a black rectangle. Qualitative figures are ok, but still a better visualization of the different regimes has to be achieved. If not there is no real point to show the rasters

When you use MFR and MBR in the results for the first time, there is no mention of their abbreviation and meaning.

Line 179: here you use RULE 1? say it clearly and not in between brackets in the previous sentence. also say what connectivity you establish between the layers, do you have similar rules to within layer? All this is defined in methods but you use also simply mention in the results. As I said part of the material you put in methods, it is indeed relevant to move it in the result, since it builds your model. Strictly speaking only the LIF neurons and parametres used are for the methodological section. Topology within layer and inter-layer connectivity is your model and is needed a clear explanation in the result section.

Figure 2: is this only 1 layer (2D) vs how many layers? details are missing in results and legend to make it clear.

Line 185-188: “No significant variations were observed in the shape of the network bursts, both in the rise (Fig 2F, left) and in the decay (Fig 2F, right) phases, suggesting that the addition of the third dimension itself is not a sufficient condition to shape the collective behavior of the neuronal network.” I mean you just show that 3D networks behave collectively differently in terms of NBBD, so now you contradict what you just said

Line 188-191: “These results are in line with the considerations reported in [12], where

189 it was proposed that the key parameter in the modulation of the network burst dynamics is the alteration 190 of the E/I balance in the 3D connectivity due to different interacting populations (cortical and hippocampal 191 in that study).” Move this to discussion, this is confusing especially when different source circuits are mentioned (hippocampus and cortex)

Line 232-239: this is not clear. if you rely on previous measurements, then you have to better summarize the result without the need to interrupt the reading of this paper and read the paper [21] to be able to advance. how inhibitory and excitatory connections were extracted? at least a mention is needed. Also you show here results from 2D networks, not3D, so be clear because if not it is confusing

Line 236: in the literature, from this review, the match between structure and function has been more and more shifted to the modular level. so structural-functional match on single link is not really observed and relevant, because structural-functional modules emerge with no real single link match. This has to be taken into account addressed in the work when mentioning this pioneering paper.

Line 248: what is the rationale of scale free only in the readout layer? Layer 1 is scale free and others no? you do not present any real rationale for it, although I guess this is based on less physical constraints on bottom layer which is seeded separately. As I said this kind of information which related your model to the experimental model you try to reproduce, has to be presented at the beginning of the results and not just in methods

Figure 6: while only MFR and MBR are plotted here? before you plotted more metrics. it seems only for MFR in RND and MC shows some good match. in MBR there is no real good match. also specify if the bar plots represent the 25-75% percentiles of your model so it can be compared to the 25-75% percentile of the experimental data. Although it could be obvious this has to be clearly stated

Line 489-492: where are 3 configurations? one i guess is the non-restricted one but here you discuss two.

Reviewer #3: It is an interesting work which proposes a computational approach to explore different connectivity configurations able to reproduce the activity patterns recorded by MEAs in 3D cortical networks.

Concerning the % of inhibitory neurons (optimal value 20%, Fig 1), aren’t there experimental hints to pre-define the number or density of the 2 main neuron types (excitatory and inhibitory)? Furthermore, I guess the connectivity impacts a lot on the firing patterns; it is not clear how they are connected (excitatory to inhibitory, auto-inhibition, inhibitory to excitatory…)

It is not clear then if all the different firing properties are averaged across all neurons (of the bottom layer), or only excitatory

Fig 2 E, which time bin?

It could be interesting to better comment the number of layers wrt biological properties of the cortical regions (e.g thickness)

Fig 3: the config with 3 layers (in E and F) seems to differ from some of the other configurations

Fig 4 B. Slope: what does it mean? Linear fitting?

Fig 6: for the BD, isn’t there any experimental reference value?

It is not well defined the difference between the configurations Free and Rnd ; also in rows 487-488 “random and unrestricted (Free) .. and Random (rnd)” is confusing

It should be discussed a more detailed approach in modelling. If one uses multicompartmental neuron models in microcircuits, more “bottom-up” connectivity patterns could emerge (e.g. geometrical intersection among morphologies, anisotropy …); this could validate the connectivity properties emerged by the statistical rules tested here

Table 1: I_noise: is it synaptic noise? What is gl in Values column?

Are the inhibitory and excitatory neurons modelled in the same way (same parameters)?

Is the synaptic time constant the same for exc and inhib?

How and c are set?

In general, the parameter tuning is poorly explained and it seems no validation is provided

Rule 1: how the threshold (8%) is defined to implement the pruning ?

3D connectivity models: the probabilistic control over distance seems to be isotropic. It should be discussed in limitations

In general, a limitation paragraph on the model should be inserted

Some information about the propagation along layers (from in silico available data) could be added

Typo row 284: where

I think in rows 490-494, it should refer to INTER-layer connections

10 seconds to settle the simulation; why so long?

Row 549: “inter-burst event intervals maximum threshold (set at 100 ms)” not clear

**Have the authors made all data and (if applicable) computational code underlying the findings in their manuscript fully available?**

Reviewer #1: Yes

Reviewer #2: Yes

Reviewer #3: None

PLOS authors have the option to publish the peer review history of their article (what does this mean?). If published, this will include your full peer review and any attached files.

Reviewer #1: **Yes: **Jordi Sorianp

Reviewer #2: No

Reviewer #3: No
---

## [Decision Letter · Decision Letter 1]

17 Dec 2022

Dear Prof. Massobrio,

We are pleased to inform you that your manuscript 'Modeling the three-dimensional connectivity of in vitro cortical ensembles coupled to Micro-Electrode Arrays' has been provisionally accepted for publication in PLOS Computational Biology.

One of the reviewers still had a suggestion for the caption of figure 7, that you can edit in the proofing stage if you wish.

Best regards,

Daniele Marinazzo

Section Editor

PLOS Computational Biology

Daniele Marinazzo

Section Editor

PLOS Computational Biology

Reviewer's Responses to Questions

**Comments to the Authors:**

Reviewer #1: The authors made a thorough revision and addressed all my concerns. I recommend publication.

Reviewer #2: all the issues that I have raised have been addressed. the MS is now better flowing and organized, although it became more heavy (especially in the introduction) in order to fullfill all the comments of the reviewers

Reviewer #3: The Authors have addressed all of my concerns. The revised

manuscript is ready for publication

Just a minor remark on Fig 7 caption: "For all the 3D, the connectivity between the layers follows a

Gaussian distribution scaled over distance as reported in Sect. 3D connectivity" . Misleading. There are other 3D configuration with SF rule...

**Have the authors made all data and (if applicable) computational code underlying the findings in their manuscript fully available?**

Reviewer #1: Yes

Reviewer #2: Yes

Reviewer #3: None

PLOS authors have the option to publish the peer review history of their article (what does this mean?). If published, this will include your full peer review and any attached files.

Reviewer #1: **Yes: **Jordi Soriano-Fradera

Reviewer #2: No

Reviewer #3: **Yes: **Claudia Casellato

---

## [Editor Report · Acceptance letter]

7 Feb 2023

PCOMPBIOL-D-22-01081R1 

Modeling the three-dimensional connectivity of in vitro cortical ensembles coupled to Micro-Electrode Arrays

Dear Dr Massobrio,

I am pleased to inform you that your manuscript has been formally accepted for publication in PLOS Computational Biology. Your manuscript is now with our production department and you will be notified of the publication date in due course.

With kind regards,

Bernadett Koltai
